# Reconstitution of dynein transport to the microtubule plus end by kinesin

Anthony J Roberts[1,2]*, Brian S Goodman[1], Samara L Reck-Peterson[1]*

[1]Department of Cell Biology, Harvard Medical School, Boston, United States; [2]Astbury Centre for Structural Molecular Biology, School of Molecular and Cellular Biology, University of Leeds, Leeds, United Kingdom

**Abstract** Cytoplasmic dynein powers intracellular movement of cargo toward the microtubule minus end. The first step in a variety of dynein transport events is the targeting of dynein to the dynamic microtubule plus end, but the molecular mechanism underlying this spatial regulation is not understood. Here, we reconstitute dynein plus-end transport using purified proteins from *S. cerevisiae* and dissect the mechanism using single-molecule microscopy. We find that two proteins–homologs of Lis1 and Clip170–are sufficient to couple dynein to Kip2, a plus-end-directed kinesin. Dynein is transported to the plus end by Kip2, but is not a passive passenger, resisting its own plus-end-directed motion. Two microtubule-associated proteins, homologs of Clip170 and EB1, act as processivity factors for Kip2, helping it overcome dynein's intrinsic minus-end-directed motility. This reveals how a minimal system of proteins transports a molecular motor to the start of its track.

## Introduction

Cytoskeletal motor proteins transport and position a variety of macromolecules, organelles and mRNAs in the cell interior (*Vale, 2003*), but how these motors are themselves targeted to specific locations within the cell is an important unsolved question. Cytoplasmic dynein, a large and complex AAA+ motor protein (*Carter, 2013*), uses the energy from ATP hydrolysis to move cargoes toward the minus end of microtubules (typically toward the cell center). In addition, dyneins can act while anchored at the cell cortex, where they pull the microtubule network toward them (*Moore et al., 2009*). Live-cell imaging in diverse organisms reveals that, surprisingly for a minus-end-directed motor, dynein accumulates at the plus ends of microtubules that grow and shrink near the cell periphery (*Vaughan et al., 1999*; *Han et al., 2001*; *Ma and Chisholm, 2002*; *Lee et al., 2003*; *Sheeman et al., 2003*; *Lenz et al., 2006*; *Kobayashi and Murayama, 2009*). By localizing to dynamic microtubule plus ends (*Howard and Hyman, 2009*), dynein is thought to 'search-and-capture' (*Kirschner and Mitchison, 1986*) cargo molecules, before transporting them toward the minus end (*Wu et al., 2006*). The targeting of the dynein machinery to the microtubule plus end is crucial for proper dynein function in budding yeast (*Moore et al., 2009*), filamentous fungi (*Wu et al., 2006*), and a subset of metazoan cells, including mammalian neurons (*Lomakin et al., 2009*; *Lloyd et al., 2012*; *Moughamian and Holzbaur, 2012*; *Moughamian et al., 2013*). However, the molecular mechanisms that target dynein to the microtubule plus end are poorly understood.

Microtubule plus ends are dynamic binding platforms for a variety of proteins, collectively referred to as +TIPs (plus-end tracking proteins) (*Akhmanova and Steinmetz, 2008*). Previous studies have illuminated two general mechanisms by which proteins can be targeted to microtubule plus ends. First, a subset of +TIPs preferentially bind to unique structural features present at the plus end vs the body of the microtubule, and can recruit additional proteins to these sites. The prototypical example of this class of +TIP are EBs (end-binding proteins), which are involved in recruiting binding partners such as Clip170 and the dynein regulator dynactin to plus ends in metazoans (*Watson and Stephens,*

*For correspondence: anthony_roberts@hms.harvard.edu (AJR); reck-peterson@hms.harvard.edu (SLR-P)

**Competing interests:** The authors declare that no competing interests exist.

**eLife digest** Eukaryotic cells use transport systems to efficiently move materials from one location to another. Much transport in the cell interior is achieved using molecular motors, which carry cargoes along tracks called microtubules.

Unlike roads of human construction, microtubules are very dynamic. One of their ends (the 'plus' end) explores the outskirts of the cell, growing and shrinking through the addition and loss of protein building blocks. The other microtubule end (the 'minus' end) typically lies in a hub near the center of the cell.

There are two types of molecular motor that move on microtubules. Kinesin motors move toward the plus end of the microtubule, and dynein motors move in the opposite direction, toward the minus end. But if dynein only moves to the minus end of the microtubule, a problem arises: how would dynein initially reach the plus end of the microtubule and the outskirts of the cell, where it collects cargoes?

Using purified yeast proteins, Roberts et al. reveal that a group of three proteins can solve this problem by transporting dynein to the plus end of the microtubule. The proteins comprise a kinesin motor, and two additional proteins that connect the dynein motor to the kinesin. Imaging the transport process shows that the dynein motor is not a passive passenger: it is able to resist against the kinesin. However, an additional microtubule-associated protein can help the kinesin motor to win this 'tug of war', and so the protein complex—including the dynein motor—moves toward the plus end of the microtubule.

*2006*; *Bieling et al., 2007*; *Dixit et al., 2009*; *Zanic et al., 2009*; *Maurer et al., 2012*; *Moughamian et al., 2013*). Second, kinesin motor proteins can power the vectorial transport of proteins along the microtubule body to the plus end (*Bieling et al., 2007*; *Subramanian et al., 2013*). These two mechanisms (direct recruitment and vectorial transport) are not mutually exclusive, and evidence suggests that both may contribute to the plus-end targeting of dynein (*Wu et al., 2006*).

In the yeast *Saccharomyces cerevisiae*, genetic studies implicate at least three proteins in targeting dynein to the microtubule plus end: Lis1, a regulator of dynein motility (*Lee et al., 2003*; *Sheeman et al., 2003*); Bik1, a homolog of Clip170 (*Sheeman et al., 2003*; *Markus et al., 2009*); and Kip2, a plus-end-directed kinesin motor (*Carvalho et al., 2004*; *Caudron et al., 2008*; *Markus et al., 2009*). Deletion of the EB homolog (Bim1) in *S. cerevisiae* appears to have little effect on the plus-end targeting of dynein and its co-factors (*Carvalho et al., 2004*; *Caudron et al., 2008*; *Markus et al., 2011*), despite (1) Bim1 binding directly to Bik1, an essential factor for dynein plus-end targeting (*Sheeman et al., 2003*; *Blake-Hodek et al., 2010*); and (2) The homologs of Kip2, Bik1 and Bim1 functioning together as a plus-end tracking system in fission yeast (*Bieling et al., 2007*). While dynein plus-end targeting can persist in the absence of Bim1, it is unknown if Bim1 is involved in the dynein pathway in the native situation, because +TIPs often interact in an interconnected and, in some cases, redundant manner (*Caudron et al., 2008*). More generally, how a system of molecules can function together to target dynein to the plus end is not clear in any system. This problem is particularly pressing for *S. cerevisiae* dynein: a forceful minus-end-directed motor with constitutive activity in motility assays (*Reck-Peterson et al., 2006*; *Gennerich et al., 2007*).

In vitro reconstitution can provide powerful mechanistic insights into cellular processes, complementary to those derived from the complex in vivo environment (*Liu and Fletcher, 2009*). Here, we have reconstituted kinesin-driven transport of dynein to the microtubule plus end using purified proteins, allowing us to dissect the mechanism using single-molecule imaging, protein engineering and DNA origami.

## Results

### Dynein-Lis1 and Bik1-Bim1-Kip2 form interacting sub-complexes

We began by purifying the *S. cerevisiae* proteins Lis1, Bik1, Bim1 and Kip2, in addition to a well-characterized dynein motor construct (GST-dynein$_{331 kDa}$; referred to herein as 'dynein') (*Reck-Peterson et al., 2006*). This dynein construct lacks the cargo-binding tail, which is dispensable for plus-end targeting in vivo (*Markus et al., 2009*), and is dimerized by GST to yield a motor with highly-similar

motile properties to full-length dynein (*Reck-Peterson et al., 2006*). Each purified protein migrated as a single band by SDS-PAGE (*Figure 1A*) with the exception of Kip2, which migrated as a doublet. The Kip2 bands correspond to differently phosphorylated forms of the protein; after treatment with λ phosphatase, the Kip2 doublet collapsed into a single band (*Figure 1A*, inset).

To explore the interactions within this putative dynein plus-end transport machinery, we mixed the proteins in different combinations and analyzed their behavior by size-exclusion chromatography. Dynein and Lis1 co-elute in a complex, as verified by SDS-PAGE of the fractions (*Figure 1B*; *Huang et al., 2012*). We also found that Bik1, Bim1 and Kip2 co-elute in a ternary complex (*Figure 1B*), resembling the behavior of related proteins from *Schizosaccharomyces pombe* (*Bieling et al., 2007*; *Table 1*). A pair-wise mixture of Kip2 with Bim1 did not elute as a complex under the same conditions (*Figure 1—figure supplement 1*), suggesting that Bim1 binds with higher affinity to the Kip2-Bik1

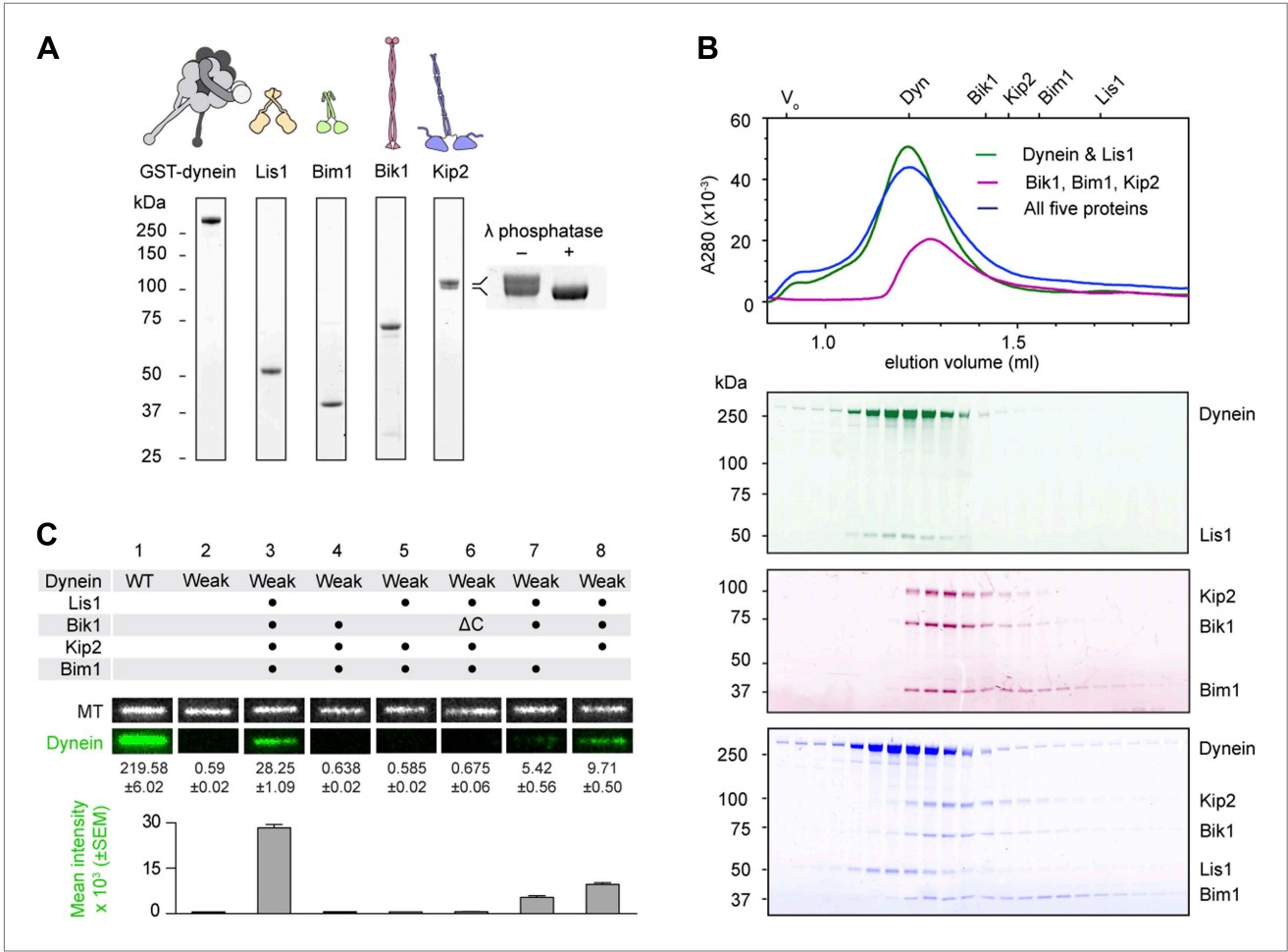

**Figure 1**. Purification and analysis of the putative dynein plus-end transport machinery. (**A**) SDS-PAGE and diagrams of purified proteins. Kip2 from *S. cerevisiae* migrates as a doublet of bands, corresponding to differently phosphorylated isoforms as verified by phosphatase treatment (right panel). (**B**) Analysis of sub-complex formation by size-exclusion chromatography. Elution volumes of the individual proteins are shown on the top axis. $V_0$: void volume. Dynein and Lis1 co-elute as a complex (green trace), as do Bik1, Bim1 and Kip2 (magenta trace). A mixture of all five proteins elutes as dynein-Lis1 and Bik1-Bim1-Kip2 sub-complexes, rather than one stable assembly (blue trace). SDS-PAGE analysis of the fractions is shown below, pseudo-colored to match the traces. (**C**) TIRF microscopy microtubule recruitment assay. Fluorescently labeled wild-type (WT) dynein or a mutant with weak microtubule affinity (K3116A, K3117A, E3122A, R3124A: Weak) were incubated with microtubules in the presence of the indicated proteins. Representative microtubule images and the corresponding dynein channel are shown in each condition. The width of each panel corresponds to 5.4 µm. Mean dynein fluorescence intensity ± SEM is shown below (N = 30–50 microtubules).

The following figure supplements are available for figure 1:

**Figure supplement 1**. Analysis of interactions between Bik1, Bim1 and Kip2 by size-exclusion chromatography.

**Table 1.** Protein homolog names

| Generic | *S. cerevisiae* | *S. pombe* |
|---------|-----------------|------------|
| Kinesin | Kip2 | Tea2 |
| Clip170 | Bik1 | Tip1 |
| EB1 | Bim1 | Mal3 |
| Lis1 | Pac1 | – |

complex. In summary, these first experiments with all five purified components demonstrate that dynein-Lis1 and Bik1-Bim1-Kip2 each form complexes that are sufficiently stable to co-elute by size-exclusion chromatography.

Given that the dynein-Lis1 and Bik1-Bim1-Kip2 sub-complexes did not stably associate in solution (*Figure 1B*), we hypothesized that these two assemblies might interact dynamically on the microtubule. To test this idea, we devised a microtubule recruitment assay. When wild-type dynein is mixed with microtubules in the absence of nucleotide, it binds strongly to the microtubule, as visualized by total internal reflection fluorescence (TIRF) microscopy in *Figure 1C* (lane 1). In order to monitor dynein recruitment to the microtubule by other proteins, we introduced four mutations into its microtubule-binding domain (*Koonce and Tikhonenko, 2000*; *Carter et al., 2008*; *Redwine et al., 2012*), which severely weakened its association with microtubules (*Figure 1C*, lane 2; see also *Figure 2—figure supplement 2*). However, the weak-binding dynein could be recruited efficiently to the microtubule by the combined presence of Lis1, Bik1, Kip2 and Bim1 (*Figure 1C*, lane 3). This dynein recruitment depended strictly on the presence of Lis1 and Bik1 (*Figure 1C*, lanes 4 and 5), consistent with the idea that Lis1 and Bik1/Clip170 interact directly (*Coquelle et al., 2002*; *Sheeman et al., 2003*; *Lansbergen et al., 2004*). To determine if the reported interaction between Lis1 and the C-terminal zinc-knuckle of Bik1/Clip170 (*Coquelle et al., 2002*; *Sheeman et al., 2003*; *Lansbergen et al., 2004*) is involved in connecting the dynein-Lis1 and Bik1-Bim1-Kip2 complexes, we purified a Bik1ΔC construct lacking the C-terminal region. The Bik1ΔC construct retained the ability to bind Kip2 and Bim1 (*Figure 1—figure supplement 1*). However, its ability to recruit the weak-dynein to the microtubule in the presence of Lis1, Kip2 and Bim1 was abolished (*Figure 1C*, lane 6). Together, these results suggest that dynein-Lis1 and Bik1-Bim1-Kip2 each form sub-complexes, which in turn interact in a manner dependent on Lis1 and the C-terminal region of Bik1.

## Reconstitution of dynein transport to the microtubule plus end with purified proteins

Having established a potential chain of connection between dynein and Kip2 with purified proteins, we next sought to visualize the emergent motile behavior of these opposite polarity motors on dynamic microtubules. We first monitored the motility of dynein and Kip2 together, but in the absence of the other proteins, on dynamic microtubules grown from stabilized seeds using three-color TIRF microscopy (*Figure 2A*). Dynein and Kip2 were labeled with different-colored fluorophores (tetramethylrhodamine and Atto647, respectively), while microtubules contained a fraction of tubulin labeled with a third color (Alexa488). In the absence of Lis1, Bik1, and Bim1, dynein moved toward the minus end of the microtubule, and Kip2 moved toward the plus end, as expected (*Figure 2B*). Notably, upon reaching the minus end of the microtubule, dynein accumulated at, and moved with, the slowly growing end of the polymer (*Figure 2B*). Dynein showed the same behavior in the absence of Kip2, indicating that dynein has the intrinsic ability to track dynamic microtubule minus ends.

Next, we explored the effect of adding the remaining factors. Strikingly, in the presence of Lis1, Bik1, Bim1 and Kip2, the large majority of dynein movements along the body of the microtubule were directed to the plus end (*Figure 2C*, lane 1) (90 ± 3%; proportion ± SE, N = 122). Moreover, on 63 ± 7% of microtubules, dynein was present at the growing plus end itself (*Figure 2—figure supplement 1*), reminiscent of dynein localization in living yeast cells (*Lee et al., 2003*; *Sheeman et al., 2003*; *Markus et al., 2009*). Minus-end-directed dynein movements were now rare (10 ± 3% of events). We observed similar behavior for the full-length dynein complex purified from *S. cerevisiae* (*Reck-Peterson et al., 2006*; *Figure 2—figure supplement 1*). Thus, we conclude that a system of four proteins (Lis1, Bik1, Bim1, and Kip2) is sufficient to strongly bias dynein's movement toward the microtubule plus end.

## Dynein can resist its transport to the plus end though its microtubule-binding domain

Dynein's plus-end-directed motion depended strictly on Kip2 (*Figure 2C*, lane 5), but was slower than that of free Kip2 (*Figure 2—figure supplement 2*), leading us to hypothesize that dynein might have the capacity to resist its own transport to the plus end. To test this model, we repeated the reconstitution

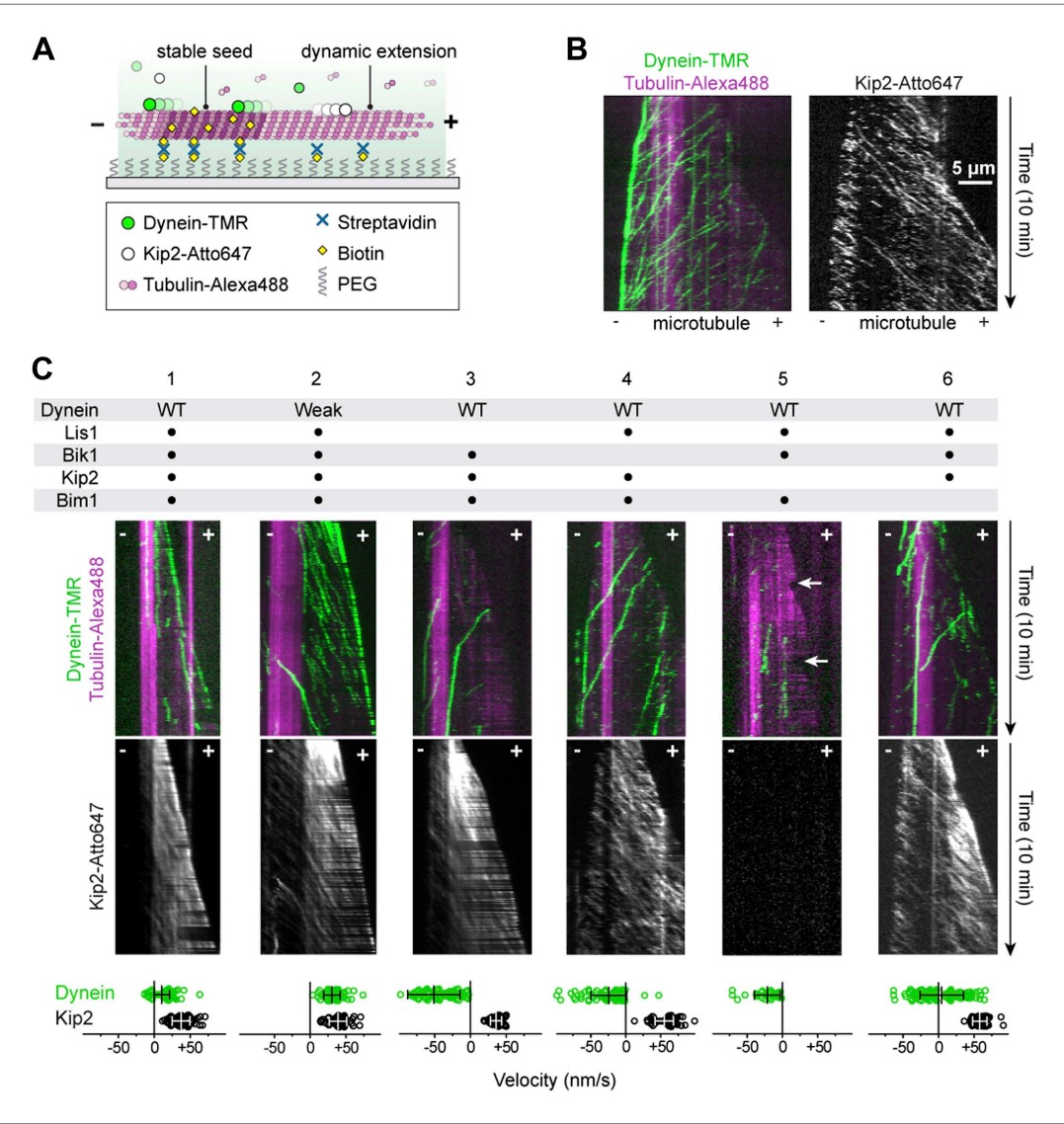

**Figure 2**. Reconstitution of dynein transport to the microtubule plus end. (**A**) Diagram of dynamic microtubule assay, after *Bieling et al. (2007)*. A bright, biotinylated, GMP-CPP-stabilized microtubule seed is attached to the coverslip via streptavidin and biotin-PEG. Dimly labeled microtubule extensions grow from this seed in the presence of tubulin and GTP. Tubulin, dynein and Kip2 are visualized using TIRF microscopy, and motor protein movement on the dynamic extensions is analyzed. (**B**) Dynein and Kip2 motility on dynamic microtubules. Kymographs of the dynein and microtubule channels are overlaid, and the Kip2 channel is shown separately for clarity. Plus (+) and minus (−) denote microtubule polarity. Dynein moves to the minus end and Kip2 moves to the plus end. Scale bar, 5 µm. (**C**) In the presence of Lis1, Bik1, Bim1 and Kip2, dynein moves to the plus end of the microtubule (lane 1). Weakening dynein's microtubule affinity via mutagenesis (*Figure 2—figure supplement 2*) increases its plus-end velocity and accumulation (lane 2). In the absence of Lis1, Bik1 or Kip2, dynein resumes minus-end-directed motion (lanes 3–5). Arrows mark microtubule catastrophe events seen in the absence of Kip2 (lane 5). When Bim1 is omitted, dynein displays both plus- and minus-end-directed movements (lane 6). Mean velocities ± SD for dynein and free Kip2 (without dynein bound) are shown at bottom (N = 36–122).

The following figure supplements are available for figure 2:

**Figure supplement 1**. Examples of plus-end-directed motion of GST-Dynein$_{331\ kDa}$ and full-length dynein.

**Figure supplement 2**. Weakening dynein's microtubule affinity increases the velocity of its plus-end-directed transport.

*Figure 2. Continued on next page*

*Figure 2. Continued*

**Figure supplement 3**. Kymographs showing colocalized, plus-end-directed runs of weak dynein and Kip2 in the presence of Bik1 and Lis1.

using the dynein mutant engineered to have weak affinity for the microtubule. Strikingly, the plus-end velocity of the weak-binding dynein was increased relative to the wild-type protein, becoming close to that of Kip2 (*Figure 2C*, *Figure 2—figure supplement 2*). Moreover, the weak-binding dynein also showed strong accumulation at the plus end of the microtubule (*Figure 2C*, lane 2). These results indicate that wild-type dynein is not a passive passenger during its plus-end-directed transport. Instead, the data support a model in which wild-type dynein and Kip2 have the potential to engage simultaneously with the microtubule in a 'tug-of-war'-like mechanism.

If dynein's plus-end-directed movement is driven directly by Kip2, these two species should colocalize and move together on the microtubule. As expected from our biochemical data (*Figure 1B*), binding events between the dynein and Kip2 subcomplexes were extremely rare when both species were diluted to the sub-nanomolar concentrations required for single-molecule imaging. However, by exploiting the weak-binding dynein construct that does not bind appreciably to the microtubule at low nanomolar concentrations (*Figure 1C*, lane 2), we were able to observe clear co-localized movements of dynein with Kip2 in the presence of Bik1 and Lis1 (*Figure 2—figure supplement 3*). Notably, the dynein construct frequently localized with Kip2 midway through a run, and dissociated before its end, consistent with transient binding as indicated by our biochemical experiments (*Figure 1B*). These observations suggest that the movement of dynein toward the plus end is driven directly by Kip2.

## Lis1 and Bik1 couple dynein to Kip2, and Bim1 regulates the efficiency of dynein plus-end transport

To probe the roles played by individual components in the plus-end-directed transport of dynein, we performed dropout experiments in which proteins were selectively omitted from the reconstitutions (*Figure 2C*). Lis1 and Bik1 were critical for dynein's plus-end-directed movement (*Figure 2C*, lanes 3 and 4). In their absence, dynein moved to the minus end of the microtubule, despite the continued movement of Kip2 to the plus end. These results indicate that Lis1 and Bik1 are required to connect dynein to Kip2. Omitting Kip2 abolished dynein's plus-end-directed movement, showing that the pathway we have reconstituted is kinesin dependent (*Figure 2C*, lane 5). Interestingly, removal of Kip2 also increased the frequency of microtubule catastrophes (the conversion from growth to shrinkage; *Figure 2C*, arrows), consistent with the short microtubule phenotype caused by Kip2 deletion in vivo (*Cottingham and Hoyt, 1997*). Finally, we observed an unexpected influence of Bim1 (*Figure 2C*, lane 6). In the absence of Bim1, dynein could still be transported to the plus end, but the fraction of plus- vs minus-end-directed dynein movements was roughly balanced (47 vs 53 ± 5%, respectively). Thus, Lis1, Bik1 and Kip2 constitute the minimal machinery for transporting dynein to the plus end, and Bim1 can regulate this machinery to increase the fraction of dynein movements that are directed to the plus end.

## Bik1 and Bim1 enhance Kip2 processivity

Because Bim1 forms a ternary complex with Bik1 and Kip2 (*Figure 1B*), we suspected that the influence of Bim1 on dynein's plus-end transport might be exerted through these proteins. Furthermore, analysis of the Kip2 amino acid sequence revealed that there is an SxIP motif within an extension N-terminal to the kinesin motor domain (*Figure 3A*). The SxIP motif is a signature sequence that interacts with EB-family proteins such as Bim1 (*Honnappa et al., 2009*), suggesting that Bim1 may interact with Kip2 through this motif. Bim1 also binds to the Cap-Gly domain of Bik1 (*Blake-Hodek et al., 2010*) and the microtubule lattice (*Zimniak et al., 2009*), while Bik1's coiled coil is thought to interact with Kip2 (*Newman et al., 2000*; *Carvalho et al., 2004*). These observations suggest that a network of interactions exists within the Bik1-Bim1-Kip2 complex (*Figure 3A*), consistent with our pairwise binding analysis (*Figure 1—figure supplement 1*). Therefore, we wanted to test if Bik1 and Bim1 could promote Kip2's microtubule interactions by providing additional, transient microtubule-binding sites (*Figure 3A*).

To directly visualize the impact of Bik1 and Bim1 on Kip2's motility, we fluorescently labeled Kip2 with Atto647 and tracked the movement of single molecules along Taxol-stabilized microtubules

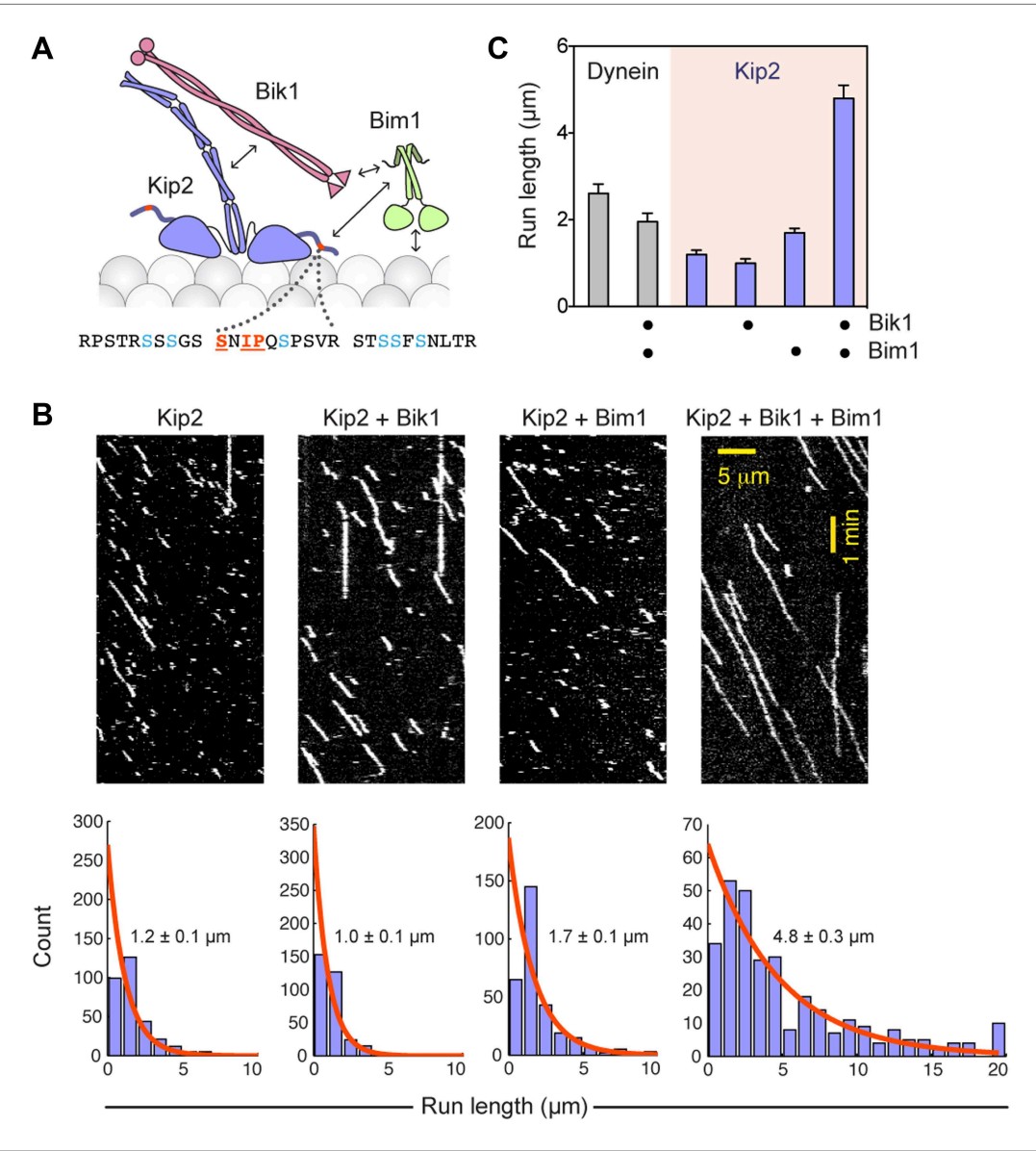

**Figure 3**. Bik1 and Bim1 are Kip2 processivity factors. (**A**) Top: diagram of Kip2, Bik1 and Bim1 domain structure. Arrows indicate reported and putative interactions (see main text for details). Bottom: Kip2 contains an SxIP motif (orange) within an N-terminal extension to its kinesin motor domain. The SxIP motif is flanked by serine residues (blue) that can be phosphorylated (*Holt et al., 2009*; *Bodenmiller et al., 2010*). (**B**) Top: kymographs showing the movement of Kip2-Atto647 along Taxol-stabilized microtubules in the absence and presence of Bik1 and Bim1. Scale bars, 1 min and 5 μm. Bottom: histograms showing the distribution of Kip2 run lengths in each condition. Orange curves show single exponential fits, based on the cumulative distribution function of each dataset. Average run lengths were determined from the decay constant. Standard errors were determined by bootstrapping, with each dataset resampled 200 times. (**C**) Bar chart of run length ± SE of dynein and Kip2 toward the minus and plus end of the microtubule, respectively, in the indicated conditions (N = 233–331).

The following figure supplements are available for figure 3:

**Figure supplement 1**. While increasing Kip2 processivity, Bik1 and Bim1 confer a small reduction in Kip2 velocity and do not change Kip2 copy number.

**Figure supplement 2**. Bik1 and Bim1 increase Kip2's microtubule on-rate and decrease its off-rate.

(*Figure 3B*). In the absence of regulators, the average distance traveled by Kip2 per microtubule encounter was 1.2 ± 0.1 µm (*Figure 3C*). The addition of Bik1 or Bim1 individually had minor effects on Kip2 run length. However, in the presence of both Bik1 and Bim1, the run length of Kip2 increased fourfold, to 4.8 ± 0.3 µm (*Figure 3C*). Concomitantly, there was a small decrease in Kip2 velocity, while the Kip2 copy number (judged by fluorescence intensity per spot) was unchanged (*Figure 3—figure supplement 1*). These effects were specific to Kip2: Bik1 and Bim1 had minimal effects on dynein motility (*Figure 3—figure supplement 1*). Analysis of Kip2 landing events and run durations revealed that Bik1 and Bim1 increase the on-rate and reduce the off-rate of Kip2's microtubule encounters (*Figure 3—figure supplement 2*), indicating that these factors augment Kip2's microtubule affinity. In summary, we conclude that Bik1 and Bim1 together act as processivity factors for Kip2.

## The outcome of a tug-of-war between dynein and Kip2 is tuned by Bim1 and Bik1

Finally, we sought to investigate if regulation of Kip2's processivity could affect its ability to overcome dynein in a tug-of-war. Our reconstitutions show that Bik1 is required to connect dynein to Kip2: thus, any additional role of Bik1 in regulating the tug-of-war was obscured. Therefore, we used DNA origami (*Shih and Lin, 2010*) to couple dynein and Kip2 directly, independent of any other proteins (*Figure 4A*). Dynein and Kip2 were attached at opposite ends of a 225 nm long DNA structure comprising a 12-helix bundle, which we call the 'chassis' (*Derr et al., 2012*). Under this regime, the large majority (80 ± 2%; mean ± SEM) of dynein/Kip2-chassis movements on the microtubule were in the minus-end (dynein) direction, as visualized by the incorporation of fluorescently labeled DNA strands (*Figure 4B,C*). Velocity analysis indicated that the chassis-coupled motors were in a tug-of-war: both minus- and plus-end-directed runs exhibited reduced velocity compared to single-motor chassis (*Figure 4—figure supplement 1*).

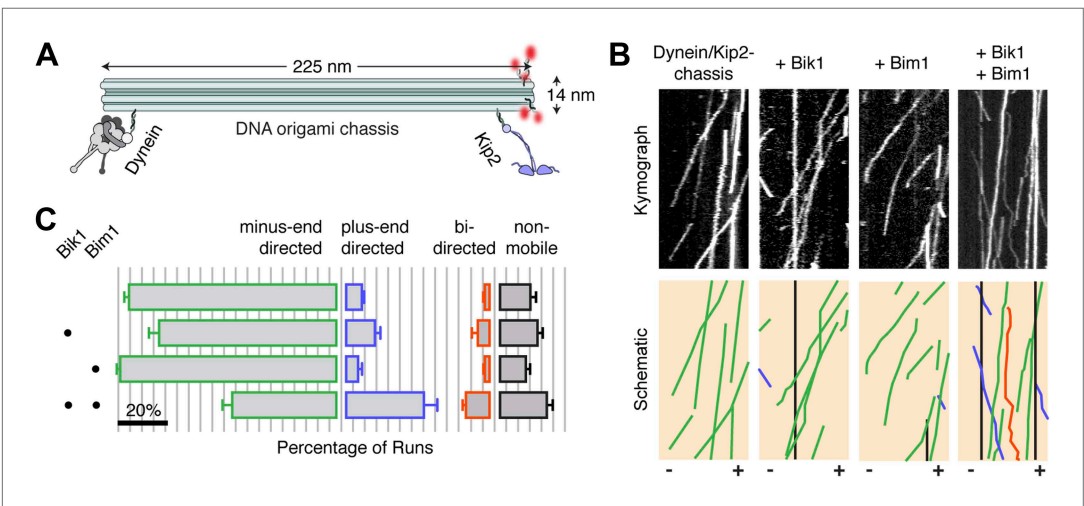

**Figure 4**. Bik1 and Bim1 regulate the outcome of a tug-of-war between dynein and Kip2. (**A**) Coupling dynein and Kip2 to a DNA origami 'chassis' allows them to be pitted directly against each other in a tug-of-war. Single-stranded DNA oligonucleotides were attached to dynein and Kip2 via SNAP tags at their N- or C-terminus, respectively (*Derr et al., 2012*). These DNA oligonucleotides base pair with complementary sequences extending from the chassis. (**B**) Top: kymographs of dynein/Kip2-chassis motility on Taxol-stabilized microtubules in the presence and absence of Bik1 and Bim1. Bottom: Color-coded schematics, highlighting runs that are minus-end-directed (green), plus-end-directed (purple), bi-directed (orange) or non-mobile (black). (**C**) Quantification of the different types of dynein/Kip2-chassis behavior in the indicated conditions, expressed as the average percentage ± SEM (N = 3 separate dynein/Kip2-chassis assembly reactions, with 306–353 runs analyzed in each case). The addition of Bik1 and Bim1 causes the fraction of plus-end-directed runs to increase relative to dynein/Kip2-chassis alone (p < 0.05; Student's t test). Scale bar, 20%.

The following figure supplements are available for figure 4:

**Figure supplement 1**. Dynein and Kip2 engage in a tug-of-war when coupled via the DNA origami chassis.

The DNA chassis allowed us to directly determine if Bik1 and Bim1 influence the outcome of the tug-of-war. The addition of either Bik1 or Bim1 alone had a modest effect on the motility of the dynein/Kip2-chassis complex: the fraction of minus- and plus-end-directed runs was changed little (*Figure 4C*). In contrast, inclusion of Bik1 and Bim1 together caused the fraction of plus-end-directed runs to markedly increase (*Figure 4C*). Under these conditions, the dynein/Kip2-chassis also underwent switches in direction during a run (*Figure 4B*; orange trace; 10 ± 1% of events). These results reveal that regulatory proteins can tune the outcome of a tug-of-war between dynein and the kinesin Kip2.

## Discussion

Through in vitro reconstitution, protein engineering and DNA origami, we have elucidated a mechanism that spatially targets dynein toward the microtubule plus end—the start of its track (*Figure 5*). The essence of the mechanism involves coupling dynein's motor domain to a plus-end-directed kinesin, Kip2. By reconstituting the functional coupling between these opposite-polarity motors for the first time, we show that two proteins, Lis1 and Bik1, are sufficient to connect dynein and Kip2. This connection is capable of bearing load on the microtubule, but is not stable enough to be observed by size-exclusion chromatography. As is the case for interactions found in other self-organizing systems (*Kirschner et al., 2000*), a high off-rate between dynein and Kip2 might allow for dynamic regulation, for example enabling dynein to readily detach from Kip2 following targeting. Indeed, in vivo, the timescale of dynein 'offloading' from the microtubule plus end onto its receptor on the cell cortex is on the order of seconds (*Markus and Lee, 2011*).

Our experiments show that dynein has the potential to resist its plus-end transport through its microtubule-binding domain. We also find that regulatory factors can tune the efficiency with which Kip2 'wins' a tug-of-war against dynein. Bik1 and Bim1 can promote Kip2's microtubule interactions and help this kinesin to overcome dynein's intrinsic minus-end-directed motility. The effect of these factors is recapitulated when dynein and Kip2 are connected synthetically using DNA origami, suggesting that it is mechanical in its basis and not contingent on a specific coupling geometry. Moreover, our biochemical demonstration of Kip2 phosphorylation suggests that additional layers of regulation may exist. Specifically, the addition and removal of phosphate groups on the N-terminal extension of Kip2's motor domain has the potential to influence its interaction with Bim1 (*Figure 3A*; *Honnappa et al., 2009*), as well as the negatively charged microtubule surface, both of which could impact Kip2 motility.

It remains to be seen if additional factors mitigate resistance from dynein during its plus-end transport in vivo, akin to the mutations in dynein's microtubule-binding domain that we found to enhance plus-end targeting in vitro. For example, dynein's tail domain is crucial for events downstream of plus-end targeting (*Markus et al., 2009*), but it is conceivable that the tail and its associated factors regulate the transport

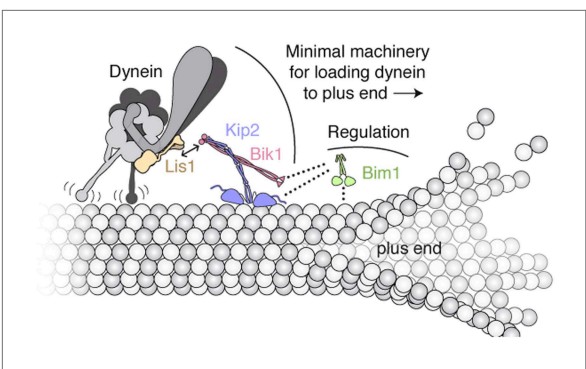

**Figure 5**. Model of the minimal machinery for transporting dynein to the microtubule plus end. Based on results from this study and earlier work. Dynein is connected to the plus-end-directed motor Kip2 by Lis1 and Bik1. Specifically, Lis1's ß-propeller domain binds to the AAA4 subdomain of dynein's motor domain (*Huang et al., 2012*). The dynein/Lis1 complex interacts dynamically with the C-terminal zinc-knuckle domain of Bik1/Clip170 (*Coquelle et al., 2002*; *Sheeman et al., 2003*; *Lansbergen et al., 2004*; *Markus et al., 2011*) as indicated by the double arrow. Bik1's coiled coil is posited to interact with Kip2 (*Newman et al., 2000*; *Carvalho et al., 2004*). Bim1 is not strictly required for dynein's plus-end transport, but together with Bik1, it can enhance Kip2's processivity and help Kip2 overcome dynein's intrinsic minus-end-directed motility. As depicted with dotted lines, it is likely that the C-terminal tail of Bim1 interacts with the N-terminal Cap-Gly domain of Bik1 (*Weisbrich et al., 2007*), and the cargo-binding domain of Bim1 interacts with the SxIP motif of Kip2 (*Honnappa et al., 2009*). This arrangement would leave the calponin-homolgy domain of Bim1 (*Slep and Vale, 2007*) free to interact transiently with the microtubule lattice, and thus promote Kip2's microtubule association. A single copy of each molecule is shown for clarity; interplay with the numerous other + TIPs in *S. cerevisiae* (not shown) is expected.

process as well. Once dynein reaches the plus end, the high local concentration of Lis1 at plus ends in *S. cerevisiae* (*Markus et al., 2011*) may help retain dynein by prolonging its microtubule attachments (*Huang et al., 2012*; *McKenney et al., 2010*; *Yamada et al., 2008*). Nudel, a Lis1 and dynein binding partner, may also promote dynein plus-end targeting by tethering Lis1 to dynein (*Li et al., 2005*; *McKenney et al., 2010*; *Wang and Zheng, 2011*; *Zylkiewicz et al., 2011*; *Huang et al., 2012*). Notably, genetic studies indicate there is a second pathway for targeting dynein directly from the cytoplasm to the plus end that does not require Kip2 (*Caudron et al., 2008*; *Markus et al., 2009*). Overlap between these pathways in vivo may explain why Bim1's capacity to regulate dynein plus-end transport was previously hidden. In summary, these results provide a vivid example of how the interplay between dynein, kinesin and microtubule-associated proteins can give rise to a coherent behavior: the recycling of a molecular motor to the start of its track.

## Materials and methods

### Protein purification

### Dynein, Lis1 and Kip2

*S. cerevisiae* strains used for the purification of dynein, Lis1 and Kip2 are listed in *Table 2*. Purification of the dimeric dynein motor domain construct (GST-dynein$_{331\ kDa}$) and full-length dynein, and labeling with tetramethylrhodamine and DNA oligonucleotides have been described (*Reck-Peterson et al., 2006*; *Derr et al., 2012*; *Qiu et al., 2012*). Lis1 was also purified as described (*Huang et al., 2012*). For Kip2, the genomic *kip2* locus was modified using homologous recombination to add an inducible *Gal1* promoter, an N-terminal His8-ZZ tag and Tev cleavage site, and a C-terminal SNAP tag. Expression of the Kip2 construct was induced by shifting cells from YP-raffinose media to YP-galactose and culturing for a further 16 hr at 30°C. Harvested cell paste was frozen in liquid nitrogen. Kip2 was typically affinity purified via its ZZ-tag as described for dynein (*Reck-Peterson et al., 2006*). For fluorescent labeling of the SNAP tag on Kip2, benzyl-guanine (BG) conjugated Atto647N dye (NEB; Ipswich, MA) was added to a final concentration of 10–30 µM while the Kip2 was bound to IgG-sepharose beads and the reaction was incubated for 20 min at ambient temperature. DNA labeling of Kip2 was carried out in the same way using 60 µM BG-oligonucleotide. After returning the mixture to 4°C, unbound

**Table 2.** Yeast strains used in this study

| Strain | Genotype | Used to purify | Source |
|---|---|---|---|
| RPY753 | $P_{GAL1}$-ZZ-Tev-GFP-3xHA-GST-DYN1$_{331kDa}$-gs-DHA-Kan$^R$; pac1Δ::klURA3; ndl1Δ::cgLEU2 | GST-dynein$_{331\ kDa}$ | (*Huang et al., 2012*) |
| RPY780 | ZZ-Tev-3xHA-DYN1-gs-DHA-Kan$^R$ nip100Δ pac1Δ::Hygro$^R$ ndl1Δ::cgLEU2 | Full-length dynein | (*Huang et al., 2012*) |
| RPY816 | $P_{GAL1}$-8HIS-ZZ-Tev-PAC1; dyn1D::cgLEU2; ndl1D::HPH | Lis1 | Julie Huang |
| RPY1084 | $P_{GAL1}$-ZZ-Tev-GFP-3xHA-SNAP-gs-GST-DYN1$_{331kDa}$-gs-DHA-Kan$^R$ | Dynein for DNA labeling | (*Derr et al., 2012*) |
| RPY1099 | $P_{GAL1}$-8HIS-ZZ-Tev-KIP2-g-FLAG-ga-SNAP–Kan$^R$ | Kip2 | This work |
| RPY1235 | $P_{GAL1}$-ZZ-Tev-GFP-3xHA-GST-DYN1$^{E3197K}$-gs-DHA-Kan$^R$ | Polarity marker dynein for chassis experiments | (*Redwine et al., 2012*) |
| RPY1536 | $P_{GAL1}$-ZZ-Tev-GFP-3xHA-GST-DYN1$^{K3116A,\ K3117A,\ E3122A,\ R3124A}$-gs-DHA-Kan$^R$ | Weak dynein | This work |

All strains were made in the W303a background (*MATa his3-11,15 ura3-1 leu2-3,112 ade2-1 trp1-1*) with genes encoding the proteases Pep4 and Prb1 deleted (*prb1Δ; pep4Δ::HIS5*). DHA and SNAP refer to the HaloTag (Promega) and SNAP-tag (NEB), respectively. *Tev* indicates a Tev protease cleavage site. $P_{GAL1}$ denotes the galactose promoter, which was used to induce protein expression. Amino acid spacers are indicated by *g* (glycine), *ga* (glycine–alanine), and *gs* (glycine–serine).

dye/DNA was removed by two washes with TEV buffer (50 mM Tris–HCl [pH 8.0], 150 mM potassium acetate, 2 mM magnesium acetate, 1 mM EGTA, 10% glycerol, 5 mM β-mercaptoethanol, 1 mM PMSF, and 0.1 mM Mg-ATP). Kip2 was released from beads via incubation with TEV protease for 1 hr at 16°C, resulting in cleavage from the His8-ZZ tag.

For size-exclusion chromatography experiments Kip2 was purified using a two-step method, which gave a higher final yield and concentration. Cells were lysed in an electric coffee grinder pre-chilled with liquid nitrogen, and resuspended in phosphate lysis buffer (final concentrations: 50 mM potassium phosphate [pH 8.0], 150 mM potassium acetate, 150 mM NaCl, 5 mM β-mercaptoethanol, 10% glycerol, 0.2% TritonX-100, 1 mM PMSF, and 0.1 mM MgATP) supplemented with 10 mM imidazole (pH 7.5). Subsequent steps were at 4°C unless indicated. The lysate was clarified by centrifugation at 264,900×$g$ for 1 hr. The supernatant was incubated with Ni-NTA agarose (Qiagen; Germantown, MD) for 1 hr, transferred into a column, washed three times with phosphate lysis buffer supplemented with 20 mM imidazole, and eluted with phosphate lysis buffer supplemented with 250 mM imidazole. The eluate was incubated with IgG sepharose beads for 1 hr, transferred into a column, and washed twice with phosphate lysis buffer and once with TEV buffer. Kip2 was released from beads via TEV cleavage as described above.

### Bik1 and Bik1ΔC

The Bik1 open-reading frame (ORF) was amplified from *S. cerevisiae* genomic DNA and inserted into the vector pKL with an N-terminal His8-ZZ tag for expression using the Baculovirus system (***Fitzgerald et al., 2006***). A Bik1ΔC construct lacking the C-terminal 40 amino acids was also constructed. Virus production and protein expression were carried out using sf21 insect cells. Proteins were purified essentially as described (***Blake-Hodek et al., 2010***). All steps were performed at 4°C. Cell pellets were resuspended in lysis buffer (50 mM Tris pH 8.5, 300 mM KCl, 5% glycerol, 10 mM imidazole, 1% NP40, 5 mM BME) and lysed using a Dounce homogenizer (10 strokes with loose plunger, 10 strokes with tight plunger). The lysate was clarified by ultracentrifugation at 183,960×$g$ for 30 min, and incubated with Ni-NTA agarose in batch for 1 hr. Following transfer into a disposable column, the agarose was washed with Buffer A (20 mM Tris pH 8.5, 500 mM KCl, 20 mM imidazole, 5 mM BME), Buffer B (20 mM Tris pH 8.5, 1 M KCl, 20 mM imidazole, 5 mM BME), Buffer A then Buffer D (20 mM Tris pH 7.5, 200 mM KCl, 5 mM BME). Finally, Bik1 was eluted with Buffer D supplemented with 300 mM imidazole and 10% (vol/vol) glycerol.

### Bim1

The Bim1 ORF was amplified from *S. cerevisiae* genomic DNA and inserted into the pDEST17 expression vector with an N-terminal His6-Strep tag and a TEV cleavage site. Expression was induced in *E. coli* (BL-21[DE3]) at OD 0.6 with 0.1 mM IPTG for 16 hr at 18°C. Cell pellets were resuspended in lysis buffer with 1 mg/ml lysozyme and lysed by sonication. Subsequent steps were at 4°C unless indicated. The lysate was clarified by ultracentrifugation at 154,980×$g$ for 30 min at 4°C, then passed twice over Strep-Tactin resin in a disposable column. The resin was washed twice with Buffer B and twice with TEV buffer. Bim1 was released from the resin via incubation with TEV protease for 1 hr at 16°C, resulting in cleavage from the His6-Strep tag.

## Protein analysis

Proteins were analyzed by SDS-PAGE on 4–12% Tris-Bis gels with Sypro Red staining (Invitrogen; Carlsbad, CA), and imaged using an ImageQuant 300 gel imaging system (Bio-Rad; Hercules, CA). Protein concentrations were determined by comparisons with standards using Bradford protein assays. All protein concentrations are expressed for the monomer, with the exception of α/β-tubulin, for which the dimer concentration is given. To verify Kip2 phosphorylation, purified Kip2 (2 μg) was incubated ±280 units of λ phosphatase (NEB) with 1 mM $MnCl_2$ in PMP NEBuffer for 30 min at 25°C. With the exception of *Figure 1A* (inset), the Kip2 used in all experiments was not treated with phosphatase.

## Size-exclusion chromatography

For size-exclusion chromatography, indicated combinations of purified protein (100 picomoles each) were pre-incubated at a concentration of 1.1 μM for 10 min at 4°C. Samples were fractionated on a Superose 6 PC 3.2/30 column using an ÄKTAmicro system (GE Healthcare; Piscataway, NJ) that had been equilibrated with gel filtration buffer (50 mM Tris–HCl [pH 8.0], 150 mM potassium

acetate, 2 mM magnesium acetate, 1 mM EGTA, 5% glycerol, and 1 mM DTT). Fractions (50 μl) were analyzed by SDS-PAGE.

## Microtubule recruitment assay

Taxol-stabilized microtubules containing 10% Alexa488-tubulin and biotin-tubulin were prepared using standard methods (http://mitchison.med.harvard.edu/protocols.html). Flow chambers for total internal reflection fluorescence (TIRF) microscopy were assembled using biotin-PEG cover glasses from MicroSurfaces, Inc. (Englewood, NJ) or prepared as described (*Bieling et al., 2010*). Chambers were incubated sequentially with the following solutions, interspersed with two washes with assay buffer (BRB80 [80 mM PIPES-KOH pH 6.8, 1 mM MgCl$_2$, 1 mM EGTA], 0.5 mg/ml casein and 1 mM DTT) with 20 μM taxol: (1) 0.5 mg/ml streptavidin in BRB80 (4 min incubation); (2) a 1:100 dilution of microtubule solution (2 min incubation). Finally, TMR-labeled dynein (2.5 nM) was added with Lis1 (50 nM), Bik1 (50 nM), Bim1 (5 nM) and Kip2 (2.5 nM) as indicated. The final reaction solution contained no nucleotide, 20 μM taxol and an oxygen scavenging system (*Yildiz et al., 2003*) in assay buffer. Alexa488-microtubules and dynein-TMR were visualized using an Olympus IX-81 TIRF microscope with a 100x 1.45 N.A. oil immersion TIRF objective and CW 491 nm and 561 nm lasers, controlled by Metamorph software (*Qiu et al., 2012*). Images were recorded with a 100 ms exposure on a back-thinned electron multiplier CCD camera, giving 159 nm/pixel at the specimen level. All conditions were imaged with identical microscope settings (491 nm laser power: 0.65 mW; 561 nm laser power: 0.35 mW; EMCCD gain: 150). Dynein fluorescence intensity on the microtubule was quantified using ImageJ. Background signal was subtracted using a rolling ball radius of 5 pixels. Intensities were determined over a 20-pixel wide line drawn perpendicular to the long axis of the microtubule, and averaged for n = 30–50 microtubules in each condition.

## Dynamic microtubule assays

Dynamic microtubule assays and visualization by TIRF microscopy were performed essentially as described (*Bieling et al., 2010*). Brightly-labeled, biotinylated microtubule seeds were polymerized by mixing Alexa488-tubulin (10 μM), biotin-tubulin (10 μM) and unlabeled tubulin (10 μM) with 0.5 mM GMP-CPP (Jena Bioscience) in BRB80 and incubating for 30 min at 37°C. Following the addition of 10 vol of BRB80, polymerized seeds were pelleted in a benchtop centrifuge (15 min at 16,100×*g*) and resuspended in a volume of BRB80 equal to the original polymerization volume. Flow chambers for TIRF microscopy were assembled using biotin-PEG cover glasses. Chambers were incubated sequentially with the following solutions, interspersed with two washes with assay buffer: (1) 1% plurionic F-127 and 5 mg/ml casein in BRB80 (8 min incubation); (2) 0.5 mg/ml streptavidin in BRB80 (4 min incubation); (3) a fresh 1:2000 dilution of microtubule seed solution (2 min incubation). The final reaction solution contained 1 mM Mg-ATP, 1 mM GTP, 0.1% methylcellulose, an oxygen scavenging system, and 15 μM tubulin (7.5% Alexa488 labeled, 92.5% unlabeled) in assay buffer. Dynein-TMR (0.04–0.3 nM), Lis1 (50 nM), Bim1 (5 nM), Bik1 (50 nM) and Kip2-Atto647 (1 nM) were added as indicated. After sealing with vacuum grease, flow chambers were imaged immediately. Three-color TIRF movies (capturing Alexa488, TMR and Atto647 fluorescence) were recorded with 3 s intervals per channel for 10 min using the microscopy setup described above and in *Qiu et al. (2012)*. Microtubule plus ends were assigned based on two criteria, which were consistent: (1) the direction of Kip2 movement and (2) the microtubule end with faster growth rate. Only motility on the dimly labeled plus-end extension of the microtubule was analyzed. Dynein velocities and directionalities were calculated from kymographs using an ImageJ macro (provided in *Supplementary file 1*). The density of Kip2 on the microtubule precluded measuring the velocity of every run in each kymograph. Hence the velocity of a subset of discrete runs was determined as an estimate of the population velocity. These velocities match closely those of Kip2 at single-molecule concentrations in equivalent conditions (*Figure 3—figure supplement 1*). In some cases, minor image drift was corrected using the ImageJ plugin *Turboreg*. Movies showing significant drift were discarded.

## Motility assays on Taxol-stabilized microtubules

Flow chambers were assembled using cover glasses (Corning no. 1½) and Taxol-stabilized microtubules containing 10% Alexa488- and biotin-tubulin were immobilized on the glass surface with Biotin-BSA and streptavidin as described (*Derr et al., 2012*; *Huang et al., 2012*). Atto647-labeled Kip2 (0.01–0.05 nM) was added with Bik1 (100 nM) and Bim1 (5 nM) as indicated, in BRB80 supplemented

with 1 mM Mg-ATP, 1 mM DTT, 20 µM taxol, 2.5 mg/ml casein, and an oxygen scavenging system. Motility assays were imaged as described (*Derr et al., 2012*; *Huang et al., 2012*). Motor velocities and run lengths were calculated from kymographs. For landing rate quantification, Kip2 was analyzed at 0.01 and 0.02 nM in the presence or absence of Bik1 (100 nM) and Bim1 (5 nM). Landing events were determined from kymographs and the rate was calculated as the number of events per micron of microtubule per nM Kip2 per minute.

For visualizing the colocalization of weak dynein with Kip2, assay chambers were prepared as for dynamic microtubule assays except taxol-stabilized microtubules were attached to the biotin-PEG surface instead of GMP-CPP stabilized seeds, and all buffers contained 20 µM taxol. The final reaction contained the weak dynein construct (2 nM), Lis1 (2 nM), Bik1 (2 nM) and Kip2-Atto647 (0.2 nM).

### DNA origami

The 12 helix-bundle DNA origami 'chassis' was made as described (*Derr et al., 2012*), by rapidly heating and slowly cooling an 8064-nucleotide, single-strand DNA (ssDNA) 'scaffold' in the presence of short ssDNA 'staples' that base-pair with the scaffold to fold it into a desired shape. The folded chassis was purified by glycerol gradient centrifugation as described (*Derr et al., 2012*). The chassis contained five TAMRA-labeled ssDNAs for fluorescent visualization. The chassis also had two projecting 'handle' sequences of ssDNA at opposite ends, termed 'A' and 'B', which were used to attach DNA-labeled dynein and Kip2 respectively. The sequences for the scaffold and all oligonucleotides are listed in *Derr et al. (2012)*. Motor-chassis complexes were assembled by incubating dynein labeled with oligo A' (complementary to handle A) and Kip2 labeled with oligo B' (complementary to handle B) with the chassis for 30 min on ice. Complex assembly was verified by agarose gel shift (*Derr et al., 2012*). Chassis motility was visualized in the presence of Bik1 (100 nM) and Bim1 (5 nM) on taxol-stabilized microtubules as indicated. To determine microtubule polarity, a low concentration (~0.1 nM) of a highly processive dynein mutant (E3197K) labeled with a different fluorophore (Atto647) was included. Chassis velocities and directionalities were determined from kymographs using ImageJ. To determine the fraction of motile events, the number of observations for a given event (e.g., the number of plus end runs) was tallied and divided by the total number of observations. Fractions are expressed as the average percentage ± SEM for three separate dynein/Kip2-chassis assembly reactions, with 306–353 runs analyzed in each case.

## Acknowledgements

We thank members of the Reck-Peterson lab for useful discussions; H Arellano-Santoyo, M Cianfrocco, M DeSantis, A Leschziner, D Pellman, K Toropova and WB Redwine for critical reading of the manuscript; T Huffaker and B Lalonde for advice on Bim1 and Bik1 purification; J Huang for initial cloning; W Shih's laboratory for DNA origami expertise; the laboratory of J Sellers for advice on insect cell culture; C Dittrich and the laboratory T Richmond for the MultiBac system; and M Cianfrocco for help with baculovirus preparation.

## Additional information

### Funding

| Funder | Grant reference number | Author |
| --- | --- | --- |
| Rita Allen Foundation | no reference number | Samara L Reck-Peterson |
| National Institutes of Health | R01GM100947 | Samara L Reck-Peterson |
| Wellcome Trust | 092436/Z/10/Z | Anthony J Roberts |

The funders had no role in study design, data collection and interpretation, or the decision to submit the work for publication.

### Author contributions

AJR, Conception and design, Acquisition of data, Analysis and interpretation of data, Drafting or revising the article; BSG, Conception and design, Acquisition of data, Analysis and interpretation of data; SLR-P, Conception and design, Analysis and interpretation of data, Drafting or revising the article

## Additional files

**Supplementary file**
• Supplementary file 1. ImageJ macro used to calculate motor velocities and run lengths from kymographs.

---

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
