## [Decision Letter]

Thank you for sending your work entitled “Mechanism of dynein targeting to the microtubule plus end by kinesin” for consideration at *eLife*. Your article has been favorably evaluated by Randy Schekman and 3 reviewers, one of whom is a member of our Board of Reviewing Editors.

The Reviewing editor and the other reviewers discussed their comments before we reached this decision, and the Reviewing editor has assembled the following comments to help you prepare a revised submission.

We believe that with the appropriate changes your work will make an excellent contribution to the field of microtubule motors and one that *eLife* will be pleased to publish. Major points to consider in the revision:

1) “Figure 1: Dynein-Lis1 and Bim1-Bik1-Kip2 form sub-complexes that interact on microtubules”. I did not find the evidence for this statement, which can also be found in modified forms in the text, sufficiently strong from the data presented in Figure 1.

The authors do not rule out a mechanism in which Bim1-Bik1-Kip2 modulate binding of dynein to the microtubule indirectly, e.g., by modulating dynein's microtubule binding activity in solution or the structure of the MT lattice. The fluorescently labelled Kip2 could be used to assess colocalization of dynein and Kip2 in the same puncta in this assay in which higher resolution images are scrutinized. This should be attempted. Higher resolution images will also allow the authors to assess if increased binding of the weak MT-binding mutant of dynein in the presence of Lis1, Kip2, Bim1 and Bik1 is correlated with increased oligomerization or dynein (analogous to the experiments presented for Kip2 in Figure S3B). This would be a welcome addition to the paper.

The data in Figure 2 are consistent with a complex formed on microtubules that consists of dynein and Kip2 and is bridged by the other proteins. However, it is not possible to see traces of motile complexes that contain fluorescent dynein and Kip2 due to the excess of Kip2 in the assays. These experiments should be supplemented with at least one protein combination (plus any appropriate controls) in which Kip2 concentration is reduced. Plus end-directed movements of dynein may be rare, but they presumably would occur. If the authors' model is correct these puncta should contain Kip2.

2) I understand that the authors need to use the weak MT binding mutant in the absence of ATP to assess the influence of the other proteins on MT association of dynein. Showing stronger evidence that the same affect also occurs for the wild-type dynein in the presence of ATP would strengthen the manuscript significantly. The authors state in the Materials and methods section: “In particular, when Lis1 and Bik1 was omitted, an elevated dynein concentration (0.3 nM) was needed to observe runs, consistent with the finding that Lis1 and Bik1 can couple dynein to Kip2 and thus can help recruit dynein onto the microtubule”. It would appear that this observation forms a nice basis for a quantitative assay to assess the influence of these factors on the association of wild-type dynein with MTs. Such an assay should be attempted.

3) Presumably the polarity of the MT in Figure 2 is assigned by assuming that the plus end is growing more rapidly in all conditions. In the kymographs shown most dynein puncta on a single microtubule are moving in the same direction. An important control would be to show that the proteins added to dynein do not affect MT dynamics such that in some MTs the minus end grows more rapidly than the plus end, thus leading to erroneous conclusions about the directionality of dynein. Although I judge this an unlikely scenario it is presumably not too challenging to assess by placing the dynamic microtubules on a glass surface coupled to an immobilized motor of known directionality (such as kinesin-1). Such a control needs only to be done for one key condition.

4) The model that Kip2's MT affinity is increased by Bik1 and Bim1 is very interesting. It would seem straightforward to test this model by assessing the frequency of MT binding events or an assay similar to that used in Figure 1. The authors may have other ideas for how to assess the influence of Bik1 and Bim1 on Kip2's MT affinity.

5) It is unclear how the velocity of Kip2 can be accurately determined from the kymographs in Figure 2. The traces are densely packed and discontinuous. A convincing account of how quantification was performed needs to be included in the methods. Perhaps adding a description of how the custom software deals with these traces would also help.

6) A major inference of the paper is that the plus end targeting of dynein has been reconstituted. However, it seems that one can be more confident that plus end-directed motility has been achieved. The authors state that dynein was present at ∼60% of microtubules of the growing plus end itself, reminiscent of dynein localization in living yeast cells. No images of wild-type dynein enrichment at the plus ends are shown so it is not clear if there is more dynein at the plus end than on the rest of the lattice. Representative Kymographs can also be shown to bolster this point.

Depending on the situation observed, the authors may wish to modify their language to talk about reconstitution of “plus end-directed movement” rather than “targeting to the plus end”.

7) There is a potential caveat that the dynein here is truncated and is a GST fusion, with the GST driving dimerisation to at least some extent. Some discussion on how potentially significant this caveat is will be useful.

---

## [Author Response]

*1) “*Figure 1*: Dynein-Lis1 and Bim1-Bik1-Kip2 form sub-complexes that interact on microtubules”. I did not find the evidence for this statement, which can also be found in modified forms in the text, sufficiently strong from the data presented in*
Figure 1.

*The authors do not rule out a mechanism in which Bim1-Bik1-Kip2 modulate binding of dynein to the microtubule indirectly, e.g., by modulating dynein's microtubule binding activity in solution or the structure of the MT lattice. […] Plus end-directed movements of dynein may be rare, but they presumably would occur. If the authors' model is correct these puncta should contain Kip2*.

We have performed the dynein-Kip2 co-localization experiment (new Figure 2—figure supplement 3) and obtained positive results. Our rationale for the design of this experiment is as follows. Our biochemical experiments suggest that dynein-Lis1 and Bik1-Bim1-Kip2 interact, but do not form a stable complex even at micromolar concentrations. Thus, when performing single-molecule co-localization experiments, we have to consider the “concentration problem” (van Oijen A.M., 2011, Curr. Opin. Biotechnol. 22: 75-80): when both species are diluted to the picomolar concentrations required for single-molecule imaging on the microtubule, the chance of observing a binding event between them is extremely small.

In reconstitution experiments with all proteins present, we used nanomolar Kip2 concentrations and picomolar dynein concentrations: conditions under which individual dynein movements can be observed, but Kip2 runs are too dense to decipher if colocalization is occurring. Conversely, if dynein is used at nanomolar concentration and Kip2 is at picomolar concentrations, dynein runs are too dense to resolve colocalization. To overcome this, we exploited the weak binding dynein mutant that does not bind appreciably to the microtubule even at nanomolar concentrations. This allowed us to use an elevated dynein concentration and a Kip2 concentration suitable for single-molecule imaging. We observed clear co-localized runs on the microtubule, suggesting that Kip2 directly drives dynein’s plus-end-directed movement.

The reviewers point out that “Bim1-Bik1-Kip2 could modulate binding of dynein to the microtubule indirectly (e.g., by modulating dynein's microtubule binding activity in solution or the structure of the MT lattice).” We have adjusted our language in the paper to acknowledge this. However, three lines of evidence argue against this possibility: 1) The weak binding dynein mutant, in which four charged amino acids have been mutated to alanine, does not bind microtubules appreciably (even in the absence of nucleotide, which corresponds to the highest affinity state in dynein’s mechanochemical cycle). It seems very unlikely that the binding of Bim1-Bik1-Kip2 to weak dynein could modulate its ability to bind microtubules, because its microtubule-binding domain is intrinsically impaired. 2) In these experiments microtubules are preformed and taxol stabilized, making lattice changes unlikely. 3) We present new data showing that removal of the C-terminal 40 amino acids of Bik1 previously reported to interact with Lis1 abolishes recruitment of weak dynein to the microtubule (new Figure 1) but has no effect on its ability to bind Kip2 and Bim1. Together with the new co-localization data, these points all indicate that Lis1 and Bik1 couple dynein to Kip2, and Kip2 drives dynein’s movement toward the plus end directly.

*2) I understand that the authors need to use the weak MT binding mutant in the absence of ATP to assess the influence of the other proteins on MT association of dynein. Showing stronger evidence that the same affect also occurs for the wild-type dynein in the presence of ATP would strengthen the manuscript significantly. The authors state in the Materials and methods section: “In particular, when Lis1 and Bik1 was omitted, an elevated dynein concentration (0.3 nM) was needed to observe runs, consistent with the finding that Lis1 and Bik1 can couple dynein to Kip2 and thus can help recruit dynein onto the microtubule”. It would appear that this observation forms a nice basis for a quantitative assay to assess the influence of these factors on the association of wild-type dynein with MTs. Such an assay should be attempted*.

Thank you for this comment, which has driven home our need to state our conclusions more clearly. We use the recruitment of the weak -binding dynein construct to the microtubule as a diagnostic for interactions between dynein and its plus-end-transport machinery (we did not mean to imply that dynein is recruited to the microtubule in the same way in the in vivo situation). We believe the observation that more dynein is required to observe a similar density of runs in the absence of Bik1 and Lis1 reflects the fact that, in the presence of Kip2, there are fewer microtubule binding sites available to dynein since Kip2 and dynein compete for the same site on the microtubule. In the presence of Bik1 and Lis1, which couple dynein to Kip2, the need for dynein to bind the microtubule directly is bypassed, and lower dynein concentrations are required to observe binding events.

Based on this logic, we think the outcome of the suggested assay would depend critically on the Kip2 concentration used and its interpretation with respect to the situation in vivo would not be straightforward, so we would be unable to make any firm additional conclusions based on it. We have emphasized that we used microtubule recruitment for diagnostic purposes, and modified the highlighted passage.

*3) Presumably the polarity of the MT in*
Figure 2
*is assigned by assuming that the plus end is growing more rapidly in all conditions. In the kymographs shown most dynein puncta on a single microtubule are moving in the same direction. An important control would be to show that the proteins added to dynein do not affect MT dynamics such that in some MTs the minus end grows more rapidly than the plus end, thus leading to erroneous conclusions about the directionality of dynein. Although I judge this an unlikely scenario it is presumably not too challenging to assess by placing the dynamic microtubules on a glass surface coupled to an immobilized motor of known directionality (such as kinesin-1). Such a control needs only to be done for one key condition*.

We assigned plus ends of microtubules based on two criteria: 1) the direction of Kip2 movement and 2) the microtubule end with faster growth rates. In all cases the direction of Kip2 movement correlated with moving towards the faster growing microtubule end. We have clarified this in the Materials and methods.

*4) The model that Kip2's MT affinity is increased by Bik1 and Bim1 is very interesting. It would seem straightforward to test this model by assessing the frequency of MT binding events or an assay similar to that used in*
Figure 1*. The authors may have other ideas for how to assess the influence of Bik1 and Bim1 on Kip2's MT affinity*.

We have provided further analysis of Bik1 and Bim1’s effect on Kip2’s microtubule encounters in Figure 3—figure supplement 2. These factors both increase Kip2’s landing rate and reduce its off-rate, consistent with the model that they augment Kip2’s microtubule affinity. Thank you for this suggestion.

*5) It is unclear how the velocity of Kip2 can be accurately determined from the kymographs in*
Figure 2*. The traces are densely packed and discontinuous. A convincing account of how quantification was performed needs to be included in the methods. Perhaps adding a description of how the custom software deals with these traces would also help*.

We have clarified how Kip2 velocity was measured in the materials and methods. As indicated, the density of Kip2 runs in the main reconstitution precludes measuring the velocity of every molecule in the population. Therefore, we determined the velocity of a subset of discrete runs, and used the average of these values as an estimate of the population velocity. These velocities closely resemble those derived from Kip2 at single-molecule concentrations (Figure 3—figure supplement 1), and showed exactly the same trends (e.g., in both cases omitting Bim1 or Bik1 resulted in a small increase in velocity), suggesting they are a good estimate of the population velocities. We also note than none of our conclusions rest on a quantitative comparison between Kip2 velocities in the main reconstitution. The crucial comparison is between the plus-end-directed velocity of dynein with a wild-type or weak microtubule-binding domain, for which distinct runs are obtained.

*6) A major inference of the paper is that the plus end targeting of dynein has been reconstituted. However, it seems that one can be more confident that plus end-directed motility has been achieved. The authors state that dynein was present at ∼ 60% of microtubules of the growing plus end itself, reminiscent of dynein localization in living yeast cells. No images of wild-type dynein enrichment at the plus ends are shown so it is not clear if there is more dynein at the plus end than on the rest of the lattice. Representative Kymographs can also be shown to bolster this point*.

*Depending on the situation observed, the authors may wish to modify their language to talk about reconstitution of “plus end-directed movement” rather than “targeting to the plus end”*.

We have added extra kymographs (Figure 2—figure supplement 1) showing dynein at growing microtubule plus ends. While we do observe dynein at ∼60% plus ends under these conditions, we agree with the reviewers that “plus-end-directed movement” is a good phrase to describe our results, and have modified our language accordingly.

*7) There is a potential caveat that the dynein here is truncated and is a GST fusion, with the GST driving dimerisation to at least some extent. Some discussion on how potentially significant this caveat is will be useful*.

Thank you for this suggestion. We have clarified that we used GST to dimerize dynein’s motor domain, and the resulting motor has motile properties highly similar to full-length dynein (42). We aimed to stress in our original text that additional factors, such as dynein’s tail domain, have the potential to regulate the system we have reconstituted. We now state: “It remains to be seen if additional factors mitigate resistance from dynein during its plus-end transport in vivo, akin to the mutations in dynein’s microtubule-binding domain that we found to enhance plus-end targeting in vitro. For example, dynein’s tail domain is crucial for events downstream of plus-end-targeting (35), but it is conceivable that the tail and its associated factors regulate the transport process as well.”

In addition, we have added new data showing the full reconstitution performed with the full-length dynein complex purified from S. cerevisiae. We observed similar behavior for the full-length dynein as the GST-dimerized dynein (Figure 2—figure supplement 1, new panel B).